# Oxygen Levels Affect Macrophage HIV-1 Gene Expression and Delay Resolution of Inflammation in HIV-Tg Mice

**DOI:** 10.3390/v12030277

**Published:** 2020-03-01

**Authors:** Marina Jerebtsova, Asrar Ahmad, Namita Kumari, Ornela Rutagarama, Sergei Nekhai

**Affiliations:** 1Department of Microbiology, College of Medicine, Howard University, Washington, DC 20059, USA; orutagarama@gmail.com; 2Center for Sickle Cell Disease, College of Medicine, Howard University, Washington, DC 20059, USA; asrar.ahmad@howard.edu (A.A.); namita.kumari@howard.edu (N.K.); 3Department of Medicine, College of Medicine, Howard University, Washington, DC 20059, USA

**Keywords:** HIV-transgenic mice, LPS, macrophage infiltration, trans-endothelial migration, organ-specific oxygen tension

## Abstract

While antiretroviral therapy increases the longevity of people living with HIV (PLWH), about 30% of this population suffers from three or more concurrent comorbidities, whose mechanisms are not well understood. Chronic activation and dysfunction of the immune system could be one potential cause of these comorbidities. We recently demonstrated reduced macrophage infiltration and delayed resolution of inflammation in the lungs of HIV-transgenic mice. Additionally, trans-endothelial migration of HIV-positive macrophages was reduced in vitro. Here, we analyze macrophages’ response to LPS challenge in the kidney and peritoneum of HIV-Tg mice. In contrast to the lung infiltration, renal and peritoneal macrophage infiltrations were similar in WT and HIV-Tg mice. Higher levels of HIV-1 gene expression were detected in lung macrophages compared to peritoneal macrophages. In peritoneal macrophages, HIV-1 gene expression was increased when they were cultured at 21% O_2_ compared to 5% O_2_, inversely correlating with reduced trans-endothelial migration at higher oxygen levels in vitro. The resolution of macrophage infiltration was reduced in both the lung and the peritoneal cavity of HIV-Tg mice. Taken together, our study described the organ-specific alteration of macrophage dynamics in HIV-Tg mice. The delayed resolution of macrophage infiltration might constitute a risk factor for the development of multiple comorbidities in PLWH.

## 1. Introduction

Combination antiretroviral therapy (cART) significantly improved the longevity of people living with HIV-1 (PLWH), improving immune function and decreasing opportunistic infections. However, chronic long-term HIV-1 infection is complicated by the increased rates of chronic medical conditions in aging population, including cardiovascular and neurological diseases, metabolic syndrome, and non-infectious respiratory disease, which summarily contribute to HIV-1 morbidity and mortality [1,2]. About 30% of PLWH are positive for three or more concurrent comorbidities [3]. The mechanism of multiple comorbidities in PLWH population is rather multifactorial and includes high prevalence of risk behaviors in the HIV-infected populations, low levels of persistent viremia, chronic inflammation and persistent immune abnormalities. The gut mucosa plays an important role in the pathogenesis of HIV-1 infection. During the early acute phase of infection, about 80% of HIV-infected CD4+ T cells are located in the gut mucosa [4]. The loss of intestinal T cells results in increased gut permeability and microbial translocation. Despite effective cART treatment and undetectable plasma viral load, chronic immune activation, inflammation and microbial translocation in the gut persists in PLWH. Increased levels of LPS were found in the plasma of PLWH under cART [5,6]. Microbial translocation seems to be a major mechanism of chronic immune activation in PLWH that accelerates the progression of other chronic conditions [7,8]. Monocyte/macrophage response plays a major role in the development of chronic diseases in the general population and is elevated in cases of HIV-1 infection [4].

We have recently demonstrated impaired dynamics of LPS-induced lung leukocyte infiltration and the resolution of inflammation in an HIV-transgenic (HIV-Tg) mouse model [9]. HIV-Tg mice carry non-replicating and non-infectious HIV-1 transgene with the deletion of *gag* and *pol* genes. This model has been previously used for the investigation of HIV-associated renal disease [10,11,12,13], lung inflammation [14] and HIV-associated neurocognitive disorders [15]. The expression of HIV-1 genes in HIV-Tg mice is controlled by HIV-1 endogenous LTR and is tissue-specific [16]. HIV-1 gene expression was found in monocytes, macrophages, lymphocytes, renal epithelial cells and skin fibroblasts [16,17,18]. No expression was found in lung endothelial and epithelial cells. A transgenic mouse model has been used to study the long-term effects of viral proteins on the host [19]. This model is clinically relevant for people with cART-controlled HIV-1 infections, which lack viral replication but undergo the persistent stress of viral protein exposure. In the pathogen-free animal facility, HIV-Tg mice do not develop intestinal microbial translocation. Intraperitoneal injection of *E.coli* endotoxin (LPS) was used to model gut microbial products’ translocation [9]. After LPS administration, HIV-Tg mice had significantly lower lung macrophage (Mϕ) levels compared to wild type (WT) during the early timepoints of immune system activation. In contrast, lung neutrophil (polymorphonuclear cells PMNC) levels were significantly elevated in HIV-Tg mice. Additionally, we observed the delayed resolution of both Mϕ and PMNC lung infiltration in HIV-Tg mice that resulted in increased mortality [9]. Moreover, we demonstrated that HIV-1 expression in Mϕ was associated with altered trans-endothelial Mϕ migration in vitro and in mouse model. Altered leukocytes’ migration significantly exacerbated lung injury.

Here, we extended this recent study by evaluating LPS-induced leukocytes’ response in the kidney and peritoneal cavity of HIV-Tg mice. In contrast to the lungs, there were no significant differences between HIV-Tg and WT mice in the levels of renal and peritoneal Mϕ and PMNC at the early stages of immune response. HIV-1 gene expression was higher in lung Mϕs compared to the peritoneal Mϕs. Previously, we demonstrated elevated levels of HIV-1 gene expression and replication in human Mϕ cultured at a high (21%) oxygen (O_2_) level compared to the physiological oxygen levels (3–5% O_2_) [20]. To evaluate the role of oxygen tension in the Mϕ migration, we analyzed trans-endothelial migration of Mϕs isolated from the peritoneum of HIV-Tg mice in vitro and demonstrated that migration was significantly reduced at a high atmospheric oxygen level (21% O_2_) compared to the low level (5% O_2_). Reduced rates of macrophages migration correlated with the higher levels of HIV-1 expression at 21% O_2_. The resolution of macrophage infiltration was reduced in both lung and peritoneal cavity of HIV-Tg mice.

Taken together, our study describes the organ-specific alterations in leukocytes dynamics in HIV-Tg mice. Differences in the levels of infiltrating Mϕ were associated with levels of HIV-1 gene expression in infiltrating Mϕ that correlated with organ-specific oxygen tension. Delayed resolution of leukocytes’ organ infiltration might constitute a risk factor for the development of multiple chronic diseases in PLWH. Therefore, our model might be useful for the development of novel therapeutic interventions for HIV-associated chronic diseases of multiple organs, and potentially enhance the length and quality of life for PLWH.

## 2. Materials and Methods 

### 2.1. Experimental Design

All experiments were approved by the Howard University’s Institutional Animal Care and Use Committee (protocol number IACUC-MED-14-09, approved October 2, 2018 until January 27, 2021). HIV-Tg breeding pairs were obtained from the Jackson Laboratory (Bar Harbor, ME, USA) and housed in a pathogen-free environment. HIV-Tg mice and their WT littermates (5–8 weeks old, 25–30 g) were used for study. Both male and female animals were used. Mice received an intra-peritoneal (i.p.) injection of LPS (3mg/gram of body weight, *Escherichia coli*, 0111:B4, Sigma-Aldrich, St. Louis, MO, USA) and were observed for various time periods (24 h and 72 h). Blood samples were collected through the retro-orbital venous plexus. Subsequently, mice were euthanized and lungs were collected.

### 2.2. Reagents

Unless indicated, all chemicals and enzymes were obtained from Sigma-Aldrich (St. Louis, MO, USA). 1E7-03 was synthesized in-house using a previously described synthetic scheme [21]. 1E7-03 is a tetrahydroquinoline derivative drug that inhibits HIV-1 transcription with IC_50_=2 μM and does not induce cytotoxicity at concentrations below 30 μM. It disrupted the interaction of Tat protein with host protein phosphatase 1 (PP1) [21].

### 2.3. Immunohistochemistry

Paraffin-embedded lung sections were cut at 5mm, de-paraffinized, rehydrated, and stained as previously described [22]. Section were labeled with primary rat anti-mouse F4/80 antibodies (Bio-Rad, Portland, ME, USA, 1:20 dilution), and rat anti-mouse neutrophil-specific antibodies (Abcam, Cambridge, MA, USA, 1:200 dilution) following secondary biotinylated goat anti-rat antibodies (Dako North America Inc., Santa Clara, CA, USA). The heat-induced epitope retrieval method was used for both antibodies. The sections were incubated with streptavidin-peroxidase and developed using a 3-amino-9-ethylcarbozole (AEC) kit (both from Thermo Fisher Scientific, Waltham, MA, USA). Sections were counterstained with hematoxylin (Vector Laboratories, Inc., Burlingame, CA, USA). Controls included replacing the primary antibody with equivalent concentrations of the rat non-specific serum. Images were acquired by Olympus IX51 microscope with Olympus DP72 camera (both from Olympus Corporation, Waltham, MA, USA). Quantification of lung infiltrating macrophages and neutrophils was done in ten different fields with 200x original magnification in three animals.

### 2.4. Isolation of Intra-Peritoneal Macrophages

To collect intra-peritoneal (i.p.) macrophages, mice were euthanized; 10 mL of PBS was injected i.p. and intraperitoneal lavage was collected 1 min later. Cells of intraperitoneal lavage were concentrated by centrifugation at 1500 x *g*, and resuspended in 1ml of PBS. Between three and five mice from each group were used for lavage collection.

### 2.5. Quantitative RT-PCR

Total RNA was extracted from plate-adherent lung and peritoneal macrophages using TRIzol reagent. Total RNA was column-purified using an RNA isolation kit (Thermo-Fisher Scientific, Waltham, MA, USA). First-strand cDNA was prepared from total RNA using the SuperScript II First Strand Synthesis kit (Thermo Fisher Scientific, Waltham, MA, USA). Real-time RT-PCR was performed in triplicate using SYBR Green qPCR Supermix (Roche Diagnostics, Basel, Switzerland). The HIV-1 *env* was amplified with primers: Forward 5′-TGTGTAAAATTAACCCCACTCTG, Reverse 5′- ACAACTTGTCAACTTATAGCTGGT. Mouse 18S rRNA was amplified with primers: Forward-5′-TTGACGGAAGGGCACCACCAG, Reverse 5′- CTCCTTAATGTCACGCACGA TTTC, and used for normalization.

### 2.6. Trans-Endothelial Migration Assay

Trans-endothelial migration of mouse macrophages was assessed by a modified Boyden chamber assay using peritoneal macrophages. Mice were injected i.p. with LPS (0.3 μg/mg of body weight) and peritoneal lavage was collected 48 h after injection. Cells were washed with PBS and quantified. At least three different mice were used for each experiment. Briefly, 1x10^5^ endothelial cells were placed on the polycarbonate supports in tissue culture inserts (Transwell-COL, Thermo Fisher Scientific, Waltham, MA, USA) and incubated for 2–3 days until formation of a confluent monolayer. Formation of the monolayer was monitored by measurement of trans-endothelial resistance using a voltammeter (EVOM) with ENDOHM chamber (both from World Precision Instruments, Sarasota, FL, USA). Mouse lung endothelial cells formed a confluent monolayer with resistance 67.4 ± 14.7 om/cm^2^ (*n* = 10). Human glomerular endothelial cells formed a confluent monolayer with resistance 72.43 ± 12.97 (*n* = 12). About 5 × 10^4^ mouse peritoneal macrophages were added on the endothelial cell monolayer and incubated for 24 h either at 5% or 21% oxygen. To stimulate macrophage migration into the low chamber, macrophage monocyte chemoattractant protein 1 (10 ng/mL MCP1, R&D Systems, Minneapolis, MN, USA) was added to the low chamber. Macrophages were collected from the low chamber and quantified using Trypan blue assay (Thermo Fisher Scientific, Waltham, MA, USA). Total RNA was isolated from collected macrophages and quantitative RT-PCR was performed to analyze HIV-1 *env* gene expression. To inhibit HIV-1 transcription, 1E7-03 inhibitor (1 µM) was added to the macrophages in the upper chamber.

### 2.7. Flow Cytometry

For immunostaining, 5×10^5^ cells were incubated with Fc Block-2.4G2 (BD Biosciences, Franklin Lakes, NJ, USA) antibody to block Fcγ III/II receptors in 10% goat serum for 10 min before the addition of fluorochrome-conjugated anti-mouse antibodies: anti-mouse macrophage-labelling F4/80-FITC and anti-mouse neutrophil-labelling Ly-6-PE (both are from BD Biosciences, Franklin Lakes, NJ, USA). Cells were incubated with antibodies for 30 min on ice, washed with 1 mL of PBS, resuspended in 0.5 mL of PBS and analyzed by flow cytometry on FACSVerse (BD Biosciences, Franklin Lakes, NJ, USA) instrument. Flow cytometry analysis was conducted in duplicate. The quantification of acquired flow cytometry data for peritoneal macrophages and neutrophils was performed in three mice for each treatment and for each timepoint.

### 2.8. Statistical Analysis

Statistical analysis was performed using GraphPad Prism 6 software (Graph Pad Software, San Diego, CA, USA). All data are presented as means with standard deviation. Differences between the two groups were compared using the parametric unpaired two-tailed Student’s *t*-test. When more than two groups were compared, we used one-way ANOVA followed by multiple pair-wise comparisons using the Student–Newman–Keuls method. *p* values < 0.05 were considered significant. 

## 3. Results

### 3.1. Renal and Peritoneal Leukocytes Infiltrations are Similar in HIV-Tg and WT Mice

Intra-peritoneal LPS administration (1mg/kg of body weight) induced PMNC and Mϕ infiltration in the lung, kidney (Figure 1) and peritoneal cavity (Figure 2) of both WT and HIV-Tg mice at 24 h after administration (Figure 1B, PMNC WT-PBS versus WT-LPS, *p* = 0.0008; HIV-Tg-PBS versus HIV-Tg-LPS, *p* = 0.001). No differences were found in lung PMNC levels in WT and HIV-Tg mice injected with PBS (Figure 1B, *p* = 0.5476). Higher levels of PMNC were found in the lungs of HIV-Tg mice compared to WT mice after LPS administration (Figure 1A,B, red color, *p* = 0.005). Both WT and HIV-Tg mice had similar levels of Mϕ after PBS administration (Figure 1D, *p* = 0.6018). LPS significantly induced lung Mϕ levels in WT mice (Figure 1D, *p* = 0.0069), but not in HIV-Tg mice (Figure 1D, *p* = 0.8166). The lung levels of Mϕ were significantly reduced in HIV-Tg mice after LPS administration compared to WT mice (Figure 1C,D, red color, *p* = 0.0057). PMNC and Mϕ levels were similar in WT and HIV-Tg mice after PBS injection (Figure 1F,H, PNMC, *p* = 0.4148; Mϕ, *p* = 0.2602). LPS administration induced significant renal PMNC and Mϕ infiltration in both WT and HIV-Tg mice (Figure 1E PMNC and Figure 1G Mϕ, red color). In contrast to the lung, both renal PMNC and Mϕ levels were similar in WT and HIV-Tg mice (Figure 1F,G, PMNC *p* = 0.4989; Mϕ *p* = 0.8012).

Next, we isolated peritoneal lavage and evaluated PMNC and Mϕ levels by flow cytometry (Figure 2A,B). The strategy for gate selection for all cell populations is shown in Figure 2A. Representative pictures of gates selection for PMNC and Mϕs for isotype control (Figure 2B), WT mice (Figure 2C) and HIV-Tg mice (Figure 2D) are shown. No significant differences were found in PMNC levels (Figure 2E, H, WT mice—blue color, HIV-Tg—red color, *p* = 0.21), or in Mϕ levels (Figure 2F, H, WT mice –blue color, HIV-Tg—red color, *p* = 0.73). Interestingly, HIV-Tg Mϕs consisted of two cell populations, one with low (P2 gate) and the other with high (P3 gate) expression of F4/80 (Figure 2F). 

Taken together, in striking contrast to the LPS-induced lung leukocyte infiltration, no differences were found in renal and peritoneal leukocytes’ infiltrations at 24 h after LPS administration.

### 3.2. Expression of HIV-1 Genes is Elevated in Lung Macrophages Compared to Peritoneal Macrophages

We observed significant accumulation of Mϕs in the lung capillaries in HIV-Tg mice after LPS administration (Figure 3A,B, red color, *p* = 0.0046). In contrast, we did not find Mϕ accumulation in the renal capillary of either WT or HIV-Tg mice (Figure 3C,D, black arrow, *p* = 0.6393). No macrophages accumulation was found in either WT or HIV-Tg mice injected with PBS (Figure 1B). The accumulation of macrophages in lung capillaries may be explained by impaired Mϕ trans-endothelial migration. Previously we demonstrated that trans-endothelial migration of HIV-Tg macrophages was reduced compared to WT macrophages in vitro, and inhibition of HIV-1 gene expression restored the ability of Mϕs to migrate through endothelial cells [9].

We hypothesized that the altered pattern of Mϕ infiltration in the lung compared to the peritoneal cavity and kidney was associated with higher levels of HIV-1 gene expression in the lung Mϕs. To test this hypothesis, we collected lung and peritoneal macrophages from four HIV-Tg mice, isolated total RNA and performed quantitative RT-PCR for HIV-1 *env,* as described in Materials and Methods. Quantitative RT-PCR demonstrated significantly higher levels of HIV-1 *env* expression in lung Mϕs, compared to peritoneal Mϕs isolated from the same mouse (Figure 3E, *p* = 0.0056). 

Previously, we showed that HIV-1 gene expression was higher in human macrophages cultured at higher oxygen levels (21% O_2_) compared to physiological oxygen levels (3%–5% O_2_) [20]. Oxygen tension is one of the physiological factors that vary between the lung and peritoneal cavity. The normal alveolar partial pressure of oxygen (pO2) is 90–100 mmHg [23], whereas the normal intraperitoneal pO2 is 40–50 mm Hg [24]. Renal oxygen tensions varied in different kidney regions, with pO2 as low as 10–20 mmHg in the medulla and about 50 mmHg in the renal cortex [25]. pO2 values in renal cortex were similar to the oxygen tension in the peritoneal cavity. 

To test whether high oxygen levels increase HIV-1 gene expression in murine peritoneal Mϕs, we cultured them at 21% and 5% O_2_ for 48 h in vitro. Expression of *env* was significantly higher in the macrophages incubated at 21% than those incubated at 5% O_2_ (Figure 3F, *p* = 0.001). 

Taken together, we demonstrated higher levels of HIV-1 gene expression in lung macrophages compared to peritoneal Mϕs. Moreover, levels of HIV-1 expression were nearly 5-fold higher in peritoneal Mϕs cultured at 21% O_2_ compared to the same macrophages cultured at 5% O_2_.

### 3.3. Trans-Endothelial Migration of Macrophages Isolated from HIV-Tg Mice is Reduced at High Oxygen Level in Vitro

To investigate whether oxygen levels impact trans-endothelial migration of Mϕ in vitro, we utilized a modified Boyden chamber assay, as described in Materials and Methods. A Boyden chamber consists of a lower and an upper chamber, separated by polycarbonated membrane. Murine lung endothelial cell line (MLEC) was generated as described [9]. Endothelial cells were seeded on the membrane inserts and incubated at 21% O_2_ until the monolayers were formed. Murine peritoneal Mϕs were isolated from HIV-Tg and WT mice 48 h after LPS administration and added to the upper chamber on top of the endothelial cell monolayer. We also analyzed, as a control, Mϕs chemotaxis—their migration through empty filter without the endothelial cell monolayer. To stimulate macrophage migration into the low chamber, macrophage monocyte chemoattractant protein 1 (10 ng/mL MCP1) was added to the low chamber. We conducted these experiments separately at 21% O_2_ and 5% O_2_. The Mϕs migrated to the low chamber were collected after 24 h incubation and counted, and percent of Mϕs migrated in the low chamber was calculated. Percent of migrating WT Mϕs was similar after incubation at 5% and 21% O_2_ (Figure 4A, WT-white bars, compare 5% and 21% O_2,_
*p* = 0.7989). In contrast, a significantly lower percentage of HIV-Tg Mϕ migrated to the low chamber at 21% O_2_ compared to 5% O_2_ (Figure 4A, HIV-Tg- grey bars, 5% O_2_ versus 21% O_2_, *p* = 0.002), which was also significantly lower than the percentage of WT Mϕ at 21% O_2_ (Figure 4A, HIV-Tg—grey bar, WT—white bar, at 21%, *p* = 0.0003). No significant difference was found between WT and HIV-Tg Mϕs migration at 5% O_2_ (Figure 4A, *p* = 0.13). There was no difference in the chemotaxis of WT and HIV-Tg Mϕs at 21% and 5% O_2_ (Figure 4B). 

To test whether the reduced trans-endothelial migration of HIV-Tg Mϕs at 21% O_2_ would be also observed for human endothelial cells, the migration experiments were repeated using human renal glomerular endothelial cells (HGEC). HGEC were generated in our laboratory as previously described [26]. The interaction between intercellular adhesion molecules (ICAM) and the integrin leukocyte function-associated antigen-1 (LFA-1) is crucial for the trans-endothelial migration of leukocytes. Previously, species restriction for the interaction of ICAM-1 and LFA-1 was reported with reduced recognition of human ICAM-1 by mouse LFA-1 [27]. As expected, migration of Mϕs through HGEC monolayer was reduced compared to MLEC (Figure 4, compare percentage of migrating Mϕs at 5%O_2_ through mouse (panel A) and human (panel C) endothelial cells), but the overall migration pattern remained similar for both cell types. No significant differences were found between HIV-Tg and WT Mϕ migration at 5% O_2_ (Figure 4C, HIV-Tg—grey bar, WT—white bar, *p* = 0.0945). Migration of WT Mϕs remained similar at low (5%) and high (21%) oxygen levels (Figure 4C, white bars, *p* = 0.6308). In contrast, migration of HIV-Tg Mϕs at 21% O_2_ was significantly reduced compared to the migration of WT Mϕs (Figure 4C, HIV-Tg—grey bar, WT—white bar, *p* = 0.0382) or HIV-Tg Mϕs migration at 5% O_2_ (Figure 4C, grey bars, *p* = 0.0058).

To test whether the suppression of HIV-1 expression affects Mϕ migration, we treated Mϕs with small molecule inhibitor of HIV transcription (1E7-03, 1 µM). Treatment with the inhibitor for 72 h significantly increased the trans-endothelial migration of Mϕs at 21% O_2_, through both mouse (Figure 4D MLEC, *p* = 9.4 × 10^−5^) and human (Figure 4D, HGEC, *p* = 0.0062) endothelial cell monolayers but had no effect on chemotaxis (Figure 4D, *p* = 0.6914). 

Taken together, trans-endothelial migration of HIV-Tg Mϕs was significantly reduced at 21% O_2_ compared to 5% O_2_ through both mouse and human endothelial cell monolayers. The inhibition of HIV-1 gene expression increased Mϕ migration through both cell monolayers. 

### 3.4. Dynamics of Peritoneal Leukocytes Infiltration and Resolution of Inflammation

To study the resolution of peritoneal Mϕ infiltration, we collected peritoneal lavages and performed FACS analysis 48–72 h after LPS administration (Figure 5A–C). As levels of peritoneal Mϕ were similar in WT and HIV-Tg mice 24 h after LPS administration (Figure 2) and levels of HIV-1 gene expression in peritoneal Mϕs were relatively low (Figure 3), we expected similar kinetics of Mϕs resolution for WT and HIV-Tg. Levels of WT and HIV-Tg Mϕ were similar 48 h after LPS administration (Figure 5A gate P1 and C, WT versus HIV-Tg at 48 h, *p* = 0.6979). Unexpectedly, Mϕ levels in HIV-Tg mice increased significantly at 72 h compared to 48 h (Figure 5B gate P1 and C, HIV-Tg at 48 h versus 72 h, *p* = 0.000123) and compared to WT mice (Figure 5B gate P1 and C, WT versus HIV-Tg at 72 h, *p* = 1.84×10^−5^). Mϕ levels in WT mice receded at 72 h compared to the levels at 48 h but the difference was not statistically significant (Figure C, *p* = 0.2759). Mϕs population in HIV-Tg mice consisted of two subsets that expressed low and high levels of F4/80 (Figure 5A,B, P2 gate—high level, P3 gate—low level). Interestingly, only the subset with high F4/80 expression was increased at 72 h in HIV-Tg mice (Figure 5B). The absolute number of cells in peritoneal lavage was assessed by Trypan Blue assay and was higher at 72 h in HIV-Tg mice than in WT mice (Figure 5D, *p* = 0.009). Thus, not only the percentage of Mϕs in peritoneal lavage cells was increased, but the absolute cell number and the absolute number of peritoneal Mϕ were also increased in HIV-Tg mice peritoneum (Figure 5D, *p* = 1.98 × 10^−4^).

As we observed similar levels of trans-endothelial migration for WT and HIV-Tg Mϕs at 5% O_2_ (Figure 4A), we tested the proliferation of peritoneal Mϕs when cultured at 5% O_2_ and 21% O_2_. Mϕs were cultured for 72 h at these two different oxygen levels and their numbers were quantified (Figure 5E). Numbers of both HIV-Tg and WT Mϕs were significantly increased during 5% O_2_ culture (Figure 5E, compare WT at 21% O_2_ and 5% O_2_, *p* = 0.018 and also compare HIV-Tg at 21% O_2_ and 5% O_2_, *p* = 0.0064). More robust proliferation was observed for HIV-Tg macrophages (Figure 5E, compare WT and HIV-TG at 5% O2, *p* = 0.046). Thus, elevated levels of peritoneal Mϕs at 72 h after LPS administration may be associated with the intensified proliferation of Mϕs in HIV-Tg mice.

Taken together, we demonstrated that HIV-1 gene expression in Mϕs impaired the dynamics of Mϕ infiltration in an organ-specific manner. We identified two different mechanisms that were distinct in HIV-Tg mice: altered trans-endothelial Mϕ migration at high organ oxygen tension and increased Mϕ proliferation at low organ oxygen tension. Our findings present novel mechanisms of innate immune system dysregulation in different organs in HIV-Tg mice.

## 4. Discussion

Concurrent multiple age-associated chronic organ diseases are more common in PLWH than in other groups [3]. The mechanism of these complications is in part associated with the altered function of HIV-1 infected Mϕs. The development of a variety of animal models to study the pathogenesis of HIV-associated peripheral organ diseases is beneficiary for understanding the mechanism and future development of therapeutic interventions. Here, we used HIV-Tg mice to study the dynamics of LPS-induced Mϕs infiltration in the lung, kidney and peritoneal cavity. In this model, HIV-1 genes are under the HIV-1 LTR promoter and are expressed at low levels in the Mϕs without active virus replication, thus mimicking cART-suppressed latent HIV-1 infection in humans. HIV-Tg mice are not immune-deficient and possess normal immune response to LPS [14]. Therefore, this model provides an opportunity to study the role of Mϕs in the development of peripheral organs disease independently of circulating viruses, bacteria, and immunodeficiency. HIV-1 expression in Mϕs may affect multiple functions, including phagocytosis and migration [28,29,30]. The migration of monocytes into organs depends on the interaction between ICAM and the LFA-1, which are species-specific molecules [27]. In contrast to the humanized mouse, where interaction between murine endothelial cells and human monocytes is impaired, the HIV-Tg mouse represents a model with physiologically intact monocytes and Mϕs migration. Our recent study demonstrated reduced levels of lung Mϕ accumulation 24 h after LPS administration and delayed resolution of infiltration in HIV-Tg mice compared to WT mice. We attributed these defects to the reduced trans-endothelial migration of HIV-Tg Mϕs [9]. Here, we extended this recent study and performed an analysis of Mϕs dynamics in the kidney and peritoneal cavity. Surprisingly, we found different Mϕ dynamics for different organs in HIV-Tg mice. In contrast to the lung, both WT and HIV-Tg mice had similar levels of renal and peritoneal Mϕs and PMNCs 24 h after LPS administration. HIV-Tg mice express HIV-1 genes in renal epithelial cells [16], but this expression did not affect the infiltration of renal Mϕs and PMNCs. However, the resolution of organ Mϕ infiltration was delayed in all investigated organs of HIV-Tg mice compared to WT mice. Despite the previously observed reduction in trans-endothelial HIV-Tg Mϕs migration in vitro, this did not affect HIV-Tg Mϕs infiltration in the kidney and peritoneal cavity. Thus, we suggested that the organ-specific environment might modulate the dynamics of Mϕs infiltration. Multiple organ-specific factors, including levels of inflammatory, chemo attractants, and endothelial adhesion molecules, might impact Mϕ migration [31,32]. It may also depend on the level of HIV-1 expression in Mϕs (24). In accordance with this notion, we observed significantly higher levels of HIV-1 expression in the lung Mϕs compared to the peritoneal Mϕs. Intra-peritoneal injections were used for LPS administration and LPS might affect HIV-1 expression, as LPS concentration in the peritoneal cavity may be higher than in the lungs. Previously, LPS has been shown to have a negative effect on HIV-1 infection and gene expression by inducing C-C chemokines release [33,34]. LPS also has a negative effect on HIV-1 gene expression in monocyte-derived Mϕs through the induction of interferon type I production and blockage of TLR4 signaling [35,36]. Thus, increased intra-peritoneal LPS levels compared to lung LPS, might reduce HIV-1 gene expression in peritoneal macrophages.

Oxygen tension is one of the physiological factors that vary between organs in the body and may regulate HIV-1 gene expression [20]. While the air oxygen level is 21%, it is much lower in the body, varying between 14% (lung) and 1% (bone marrow). Therefore, HIV-1 is exposed to different oxygen levels in the organism. The expression of host factors essential for HIV-1 replication depends on the oxygen level and is associated with oxygen sensor proteins HIF-1α and HIF-2. Most virology studies have been performed at 21% oxygen levels and the knowledge of the mechanistic aspects of viral infection is incomplete. HIF-1α is stabilized at low oxygen conditions and can reduce NF-κB activity in vivo and under inflammatory conditions [37]. Additionally, we have demonstrated previously that activities of CDK9/cyclin T1 (the major host kinase involving in HIV-1 transcription) and transcriptional factor Sp1 are higher at 21% O_2_ compared to 3% O_2,_ which are associated with increased levels of both Tat-activated and basal HIV-1 transcription [20]. A reduction in HIV-1 transcription was in part due to the reduction in CDK9 activity at 3% O2, as there was less Tat-associated active CDK9/cyclin T1 complex formation [20]. Thus, increased HIV-1 transcription can induce an accumulation of viral proteins and escalate viral toxicity in an organ-specific manner Oxygen tension is higher in the lung (90–100 mmHg) and relatively low in the peritoneal cavity and renal cortex (40–50 mmHg) [23,24]. Thus, both high LPS level and low oxygen tension in the peritoneal cavity might reduce HIV-1 expression in peritoneal Mϕs. However, we observed robust HIV-1 expression in peritoneal Mϕs obtained from LPS-injected mice that were cultured in vitro at 21% O_2_, suggesting that their HIV-1 expression was not impaired by LPS exposure. Thus, our in vitro experiments demonstrated that difference in the oxygen levels, rather than LPS concentrations, modulates HIV-1 expression in Mϕs. 

The study of trans-endothelial Mϕ migration in vitro demonstrated that high oxygen levels induced HIV-1 expression and reduced the migration of HIV- Mϕs. To rule out the possibility that different rates of Mϕ migrations were specific for mouse lung endothelial cells, we also used a human renal endothelial cell line. Trans-endothelial migration of HIV-Tg Mϕs was also significantly reduced for both endothelial lines exposed to 21% O_2_ compared to the migration at 5% O_2_. The increased permeability of endothelial cells in our experiments may be associated with reduced cell viability at low oxygen levels. However, we did not find significant differences in the growth kinetics and viability of endothelial cells at different oxygen concentrations. Moreover, percentages of migrating WT macrophages were similar at low and high oxygen levels. Thus, the impairment of trans-endothelial migration was associated with alterations in HIV-Tg Mϕs macrophages rather than the endothelial cells.

We demonstrated here that HIV-1 expression is activated at higher oxygen levels, impairing trans-endothelial migration. Accordingly, we observed significantly higher levels of HIV-1 expression in the lung Mϕs compared to the peritoneal Mϕs. To our knowledge, this is the first in vivo demonstration of tissue-specific levels of HIV-1 transcription in Mϕs. Moreover, different levels of HIV-1 expression correlated with the organ oxygen tension. It remains to be determined whether HIV-1 expression is increased in other organs with high pO_2,_ such as the brain or heart, and whether this also correlates with impaired Mϕ function. Interestingly brain, heart and lung complications are also among the common complications during chronic HIV infection [1,4,15]. 

Reduced trans-endothelial migration at higher oxygen tension could not explain the delayed resolution of peritoneal Mϕ infiltration in HIV-Tg mice. The second mechanism that might affect the resolution of infiltration is the increased rate of Mϕs proliferation at low oxygen levels. We showed here that low O_2_ stimulated proliferation of both WT and HIV-Tg Mϕs, with a more profound effect on HIV-Tg Mϕs. HIV-1 expression was still detectable at 5% O_2_ and might also affect cell proliferation. Thus, higher levels of peritoneal Mϕs in HIV-Tg mice might be associated with the proliferation of infiltrated Mϕs. Interestingly, higher levels of HIV-positive Mϕ accumulation were previously found in the solid tumors that typically have a hypoxic environment [30]. Thus, at least two different mechanisms are at play that may explain the abnormal Mϕs LPS-induced organ dynamics.

In addition, HIV-infected Mϕs had reduced endocytic activity even in a cART setting [29,38]. Mϕs contributes to the resolution of early inflammation by the phagocytosis of neutrophil apoptotic bodies and inflammation stimulus. The phagocytosis of neutrophil apoptotic bodies by Mϕs from people with HIV-1 infection was significantly reduced in vitro [28]. Thus, impaired Mϕs phagocytosis might also contribute to the persistence of the inflammation in both HIV-1-infected people and HIV-Tg mice.

As most studies of HIV-1 infected Mϕs have been performed in vitro, the role of organ-specific environment in HIV-1 expression and monocyte/macrophage traffic is mostly unknown. The HIV-Tg mouse model provides an opportunity to address this issue. Moreover, this model demonstrated that the level of HIV-1 gene expression in Mϕ without active viral replication affected Mϕ functions. As cART therapy does not affect HIV-1 transcription, the supplementation of cART with HIV-1 transcription inhibitors may be beneficial for the treatment or prevention of age-associated chronic diseases in PLWH. Until we develop a full HIV-1 cure, the treatment of aging population of PLWH to prevent multiple chronic diseases should be an essential part of the functional cure for chronic HIV-1 infection.

## 5. Conclusions

Organ-specific oxygen tension is a critical physiological factor that modulates Mϕs migration and proliferation in response to the inflammation, leading to the delayed resolution of inflammation. Thus, this phenomenon needs to be considered in future therapeutic approaches and the development of novel drugs to treat chronic HIV-1 disease and its complications.

## Figures and Tables

**Figure 1 viruses-12-00277-f001:**
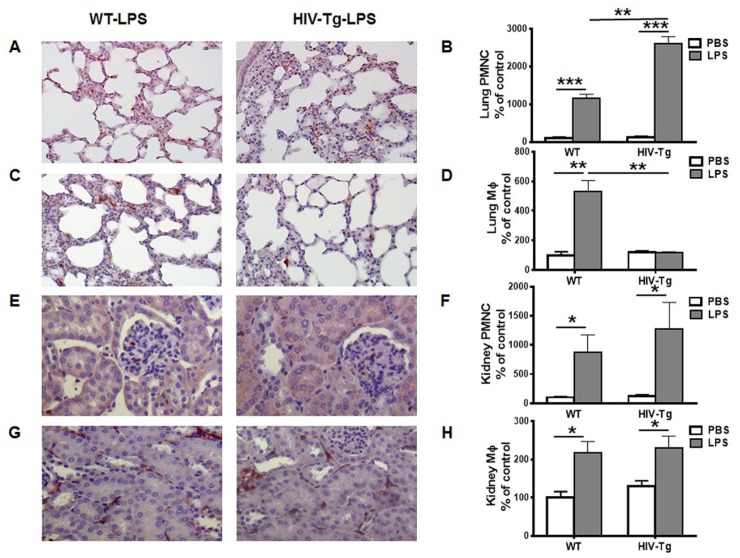
Infiltration of neutrophils is increased, and macrophages is decreased, in the lung but not the kidney of HIV-Tg mice 24 h after LPS injection. (**A**,**C**) Representative pictures of lung neutrophils (**A**) and macrophages (**C**) immunostaining (red color). Original magnification is 200X. (**E**,**G**) Representative pictures of kidney neutrophils (**E**) and macrophages (**G**) immunostaining (red color). Hematoxylin is used for counterstaining (purple nuclear staining). Original magnification is 400X. (**B**,**F**) Quantification of neutrophils (PMNC) in the lung (**B**) and the kidney (**F**). (**D**,**H**) Quantification of macrophages (Mϕ) in the lung (**D**) and kidney (**H**). Results are shown as percent of WT-PBS levels (control). Quantification is done in ten different fields with 200x original magnification in three animals in each group. Mean and standard error are shown. **p* < 0.05, ***p* < 0.01, ****p* < 0.001.

**Figure 2 viruses-12-00277-f002:**
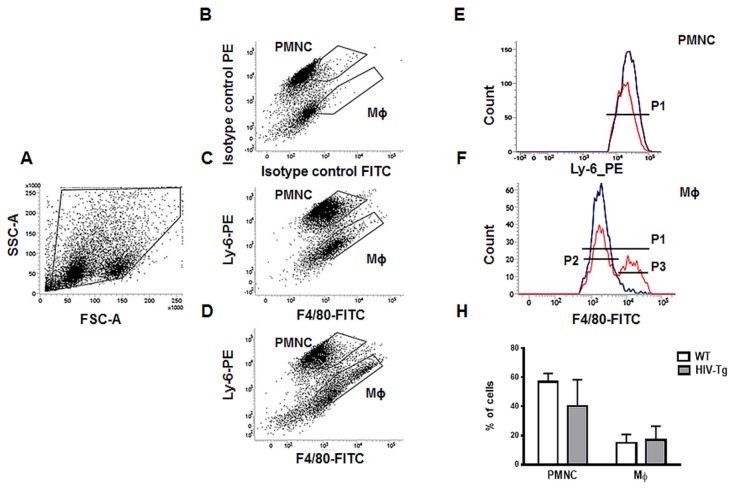
Similar levels of peritoneal neutrophils and macrophages in HIV-Tg and WT mice 24 h after LPS injection (**A**) Forward scatter versus side scatter (FCS vs. SSC) plot represents all peritoneal cell populations and gating strategy. (**B**–**D**) Flow cytometry analysis of peritoneal lavage shows neutrophil (PMNC) and macrophage (Mϕ) subsets in isotype control (panel B), WT mice (panel C) and HIV-Tg mice (panel D). (**E**,**F**) representative flow cytometry data for PMNC (**E**) and Mϕs (**F**) for WT (blue color) and HIV-Tg (red color) mice. (**H**) Quantification of flow cytometry data for peritoneal PMNC and Mϕ for P1 gate (*n* = 5). P2 is a gate for Mϕ with low level F4/80, P3 is a gate for Mϕ with high level of F4/80. Results are shown as percent of cell population. Mean and standard error are shown.

**Figure 3 viruses-12-00277-f003:**
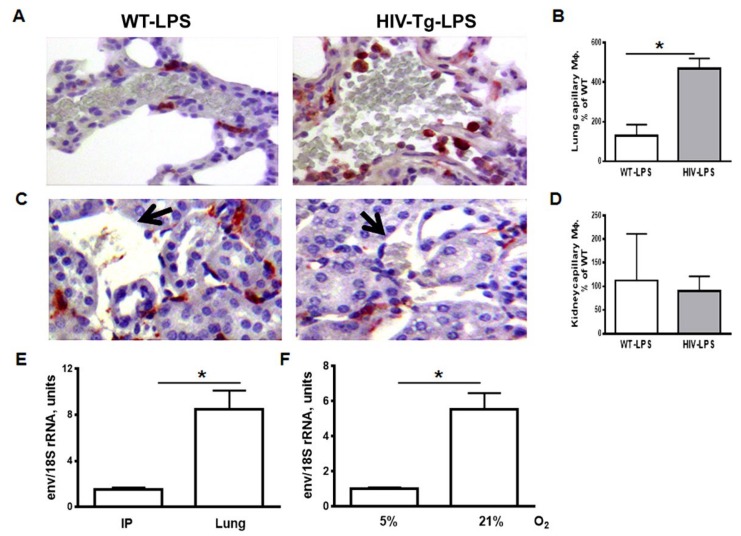
The level of capillary macrophage is increased in HIV-Tg mice at 24 h after LPS administration. (**A**,**C**) Macrophage immunostaining (red color) in the lung (panel A) and kidney (panel C) capillaries. Hematoxylin is used for counterstaining (purple nuclear staining). Original magnification is 400X. (**B**,**D**) Quantification of lung (panel B) and kidney (panel D) capillary macrophages. Results are shown as percentage of WT-LPS levels. *n* = 3. (**E**,**F**) qRT-PCR for HIV-1 *env* in peritoneal and lung Mϕs (E, *n* = 4) and in peritoneal Mϕs incubated for 72 h at 5% and 21% in vitro (F, *n* = 3). Mouse 18S rRNA was used for normalization. Mean and SD are shown. **p* < 0.05.

**Figure 4 viruses-12-00277-f004:**
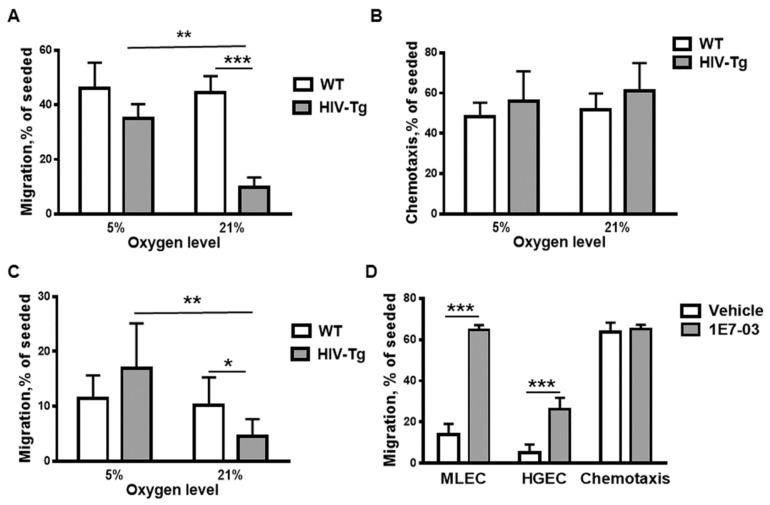
Trans-endothelial migration of HIV-Tg macrophages reduced at 21% oxygen in vitro. (**A**) Trans-endothelial migration of peritoneal macrophages isolated from WT and HIV-Tg mice through mouse lung endothelial cells at 5% and 21% oxygen. *n* = 4 (**B**) Chemotaxis of peritoneal macrophages at 5% and 21% of oxygen. *n* = 3 (**C**) Trans-endothelial migration of peritoneal macrophages isolated from WT and HIV-Tg mice through human renal endothelial cells at 5% and 21% oxygen. *n* = 6. (**D**) Treatment of peritoneal macrophages isolated from HIV-Tg mice with 1E7-03 HIV-1 transcriptional inhibitor (1 μM) significantly increases trans-endothelial migration at 21% oxygen without affecting chemotaxis. *n* = 3. Mean and SD are shown. **p* < 0.05, ***p* < 0.01, ****p* < 0.001.

**Figure 5 viruses-12-00277-f005:**
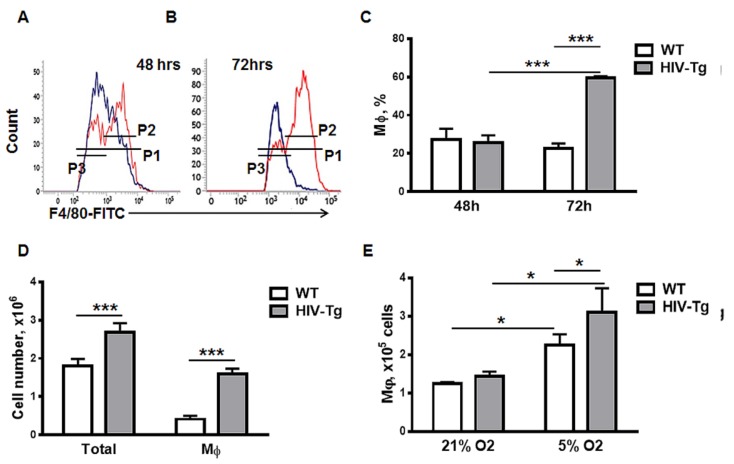
Resolution of macrophage infiltration into the peritoneal cavity is impaired in HIV-Tg mice at 72 h after LPS administration. (**A**,**B**) Representative data of flow cytometry for WT (blue) and HIV-Tg (red) macrophages at 48 (A) and 72 h (B) after LPS administration. (**C**) Quantification of peritoneal macrophages for P1 gate in WT and HIV-Tg mice at 24 h, 48 h and 72 h after LPS administration. P2 is a gate for Mϕ with low-level F4/80, P3 is a gate for Mϕ with a high level of F4/80. Results are shown as percent of peritoneal cell population. *n* = 3. (**D**) Total number of peritoneal cells and macrophages in WT and HIV-Tg mice at 48 h and 72 h after LPS administration. Results are shown as cell number. *n* = 3. (**E**) Proliferation of peritoneal macrophages isolated from WT and HIV-Tg mice at 5% and 21% oxygen in vitro. *n* = 3. Mean and SD are shown. **p* < 0.05, ****p* < 0.001.

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
