# Peer review of "Oxygen Levels Affect Macrophage HIV-1 Gene Expression and Delay Resolution of Inflammation in HIV-Tg Mice"

_viruses, 2020, doi:10.3390/v12030277_

Round 1

Reviewer 1 Report

This is a interesting paper that uses the Tg26 HIV transgenic mouse model to analyze macrophage response to LPS challenge in the kidney and peritoneum of mice. They conclude that there is a reduced macrophage infiltration and a delayed resolution of inflammation in the Tg26 mice. There are several ways that this manuscript can be strengthened and clarification is needed.

A majority of the background relies on a paper from this group that is unpublished but cited as Jerebtsova manuscript in revision or Jerebtsova manuscript in review. This citation should be removed. Statistical methods should be expanded. Some of the comparisons are of 3 or more groups and a one way ANOVA was used in general. It is unclear if these are parametric or non-parametric and what post-hoc tests were utilized. One of the main issues is that there is no HIV-TG26 alone group. In Figure 1, there are representative images of WT and Tg26 both with LPS, no example of WT alone and no HIV Tg26 alone at all. These groups should be added to the paper in all cases to determine the background of the Tg26 alone. In the Tg26 mice, all cells have the HIV transgene and not just in cells that would normally be infected with HIV. This needs to be clarified and discussed in the paper.  It needs to be clear what exactly they are measuring when they state that they find increased HIV. They are measuring env RNA only and not replication component virus.  As far as rigor is concerned it needs to be stated for each experiment and figure how many animals and how many times the study was repeated for both in vitro and in vivo and the sex of the animals.  Minor- they refer to Figure 4E but there is no Figure 4E. For the turnover/proliferation experiments in Figure 5- they are only looking at macrophage as a whole. Phenotyping should be done to strengthen the paper and demonstrate that differential populations of cells. In addition, they do not us any markers of proliferation or turnover. BrDU or other methods could be used to more actually assess cell turnover.  Minor- Please use people with HIV instead of patients with HIV.

Author Response

Specific Comment 1. A majority of the background relies on a paper from this group that is unpublished but cited as Jerebtsova manuscript in revision or Jerebtsova manuscript in review. This citation should be removed.

Response: We agree and apologize for not providing this manuscript for the reviewers. The referenced paper is now accepted for publication and available on-line. We corrected the references accordingly.

Specific Comment 2. Statistical methods should be expanded. Some of the comparisons are of 3 or more groups and a one way ANOVA was used in general. It is unclear if these are parametric or non-parametric and what post-hoc tests were utilized.

Response: We agree and update the Methods section. We indicated that parametric unpaired two-tailed Student’s t test was sued for comparison of two groups. For comparison of four groups we used ANOVA. We followed ANOVA by multiple pair-wise comparisons using the Student-Newman-Keuls test. This information was added to the Materials and Methods: Statistical analysis, section 2.8.

Specific Comment 3. One of the main issues is that there is no HIV-TG26 alone group. In Figure 1, there are representative images of WT and Tg26 both with LPS, no example of WT alone and no HIV Tg26 alone at all. These groups should be added to the paper in all cases to determine the background of the Tg26 alone.

Response: We did not have these groups as PBS has no effect. To illustrate that, we updated the Figure 1 which now shows WT and HIV-Tg groups injected with PBS as control.

Specific Comment 4. In the Tg26 mice, all cells have the HIV transgene and not just in cells that would normally be infected with HIV. This needs to be clarified and discussed in the paper.

Response: We agree that in HIV-Tg mice all cells contain HIV transgene. Because transgene contains HIV-1 endogenous LTR promoter, expression of HIV genes is cell-specific and has been widely studied (see Bruggeman L, Virology 1994, 202, (2), 940-8). HIV-1 genes were found to be expressed in monocytes, macrophages, lymphocytes, renal epithelial cells and skin fibroblasts (Bruggeman L.A. et al, 1994; Leonard J. et al, 1989, Dickie P, 2000). No expression was found in lung endothelial cells. In contrast, HIV genes were expressed in renal epithelial cells. Despite the HIV gene expression in renal cells, we did not find any differences in renal macrophages migration suggesting that this expression did not affect macrophages migration properties. Transgenic mouse model has been used to study the long-term effects of viral proteins on the host (Putatunda R et al, 2018 J Neuroinflammation). This model is clinically relevant as it resembles patients with cART-controlled HIV-1 infections, who lack viral replication but undo persistent stress of viral protein exposure. These patients display a disease phenotype that includes HIV-associated nephropathy (HIVAN), congenital cataracts, papillomatosis, and growth failure (Dickie P. AIDS Res Hum Retroviruses 2000 May 20;16(8):777-90). We added this information to the Introduction and Discussion.

Specific Comment 5. It needs to be clear what exactly they are measuring when they state that they find increased HIV. They are measuring env RNA only and not replication component virus.

Response: We agree and corrected our statements to indicate that we measured HIV RNA. HIV-Tg mouse model is replication deficient due to the deletion of viral genome and restriction of HIV-1 replication in mice.

Specific Comment 6. As far as rigor is concerned it needs to be stated for each experiment and figure how many animals and how many times the study was repeated for both in vitro and in vivo and the sex of the animals.

Response: We agree and added number of animals or experimental repeats to the figure legends. We used both male and female animals. We did not find significant differences between migration of macrophages in males and females and did not separated mice by gender in the figures. For in vitro studies we isolated peritoneal macrophages from 3-5 mice in each group and performed FACS as well as in vitro migration analysis for each mouse in duplication. This information is added to Materials and Methods, 2.4 section. 

Specific Comment 7. For the turnover/proliferation experiments in Figure 5- they are only looking at macrophage as a whole. Phenotyping should be done to strengthen the paper and demonstrate that differential populations of cells.

Response: We agree that characterization of distinct populations of the macrophages is very interesting and would increase strength of the paper. But we feel that this information will distract the readers attention from the main point of the manuscript, which is the organ specific infiltration of the macrophages in HIV-Tg mice and correlation of this migration with the organs oxygen tension. The will consider characterization of peritoneal macrophages sub-populations as a future research topic.

Specific Comment 8. In addition, they do not us any markers of proliferation or turnover. BrDU or other methods could be used to more actually assess cell turnover.   

Response: Proliferation of mouse macrophages was performed in vitro. Cells were isolated from peritoneal cavity and these cells consisted mostly of neutrophils and macrophages as shown by FACS. Neutrophils are short living cells and thus they died during 24-48 hrs of incubation. Thus we only detected the increased in macrophage number. Comparison of cell number by Trypan blue assay is scientifically sound method and we feel that we do not need to include additional method such as BrDU quantification.

Minor

  1. They refer to Figure 4E but there is no Figure 4E.

Response: We agree and corrected this typo which is now removed from the text.

  1. Please use people with HIV instead of patients with HIV.

Response: We agree and changed patients with HIV-1 to people with HIV-1.

Reviewer 2 Report

The present study on differential transmigration and retention of macrophage ad PMN in several tissues of HIV-Tg mice is interesting and well done. As the authors suggest, this model may help to address co-morbidities in well-treated cART HIV(+) patients.

It is a bit disturbing that previous manuscripts, containing probably similar experiments, are not available to the reviewers.

I have some minor remarks and questions:

126: How was confluence proven: just by microscopic view or by testing permeability to e.g. fluorescent beads of a certain size? 176-177 and 276-277: Do the authors have any idea what the biological significance is of the two macrophage populations and the selective persistence of the ‘high F4/80” population after 72 H? Was there any attempt to separate and study both populations? Is this phenomenon related to proliferation or to functional alterations ? Fig 5: Inconsistency between A-E in the figure and A-D in the legend. 337-340: All these studies were performed in human primary macrophages or cell lines. What is the relevance for the Tg mouse macrophages?

Finally, I wonder what the mechanism is of “induced HIV expression” in this model. Any idea whether a specific viral transcript is responsible or whether it is just an effect of accumulation of any viral RNA, triggering RIG/NOD/TLR with type 1 IFN effector mechanisms. Has any attempt been done to delete particular HIV genes (e.g. Tat) in order to see if the effects are the same or different? This question may be relevant as to which additional therapeutic strategies should be developed to prevent co-morbidities: block any HIV transcription or more selective inhibition of e.g. Tat? 

Author Response

Specific Comment 1. It is a bit disturbing that previous manuscripts, containing probably similar experiments, are not available to the reviewers.

Response: We agree and apologize for this omission. This manuscript is now published and referenced.

I have some minor remarks and questions:

Specific Comment 2. 126: How was confluence proven: just by microscopic view or by testing permeability to e.g. fluorescent beads of a certain size?

Response: Formation of monolayer was monitored by measurement of trans-endothelial resistance using voltammeter (EVOM) with ENDOHM chamber (both from WPI Inc.). Mouse lung endothelial cells formed confluent monolayer with resistance 67.4±14.7 om/cm2 (N=10). Human glomerular endothelial cells formed confluent monolayer with resistance 72.43±12.97 (N=12). This information was added to the Materials and Methods, 2.6 section.

Specific Comment 3. 176-177 and 276-277: Do the authors have any idea what the biological significance is of the two macrophage populations and the selective persistence of the ‘high F4/80” population after 72 H? Was there any attempt to separate and study both populations? Is this phenomenon related to proliferation or to functional alterations?

Response: Although mouse peritoneal macrophages are one of the best-studied macrophage populations, recently two subsets were described, which exhibit distinct phenotypes, functions, and origins. They are classified as large peritoneal macrophages (LPMs) and small peritoneal macrophages (SPMs). LPMs, the most abundant subset under steady state conditions, express high levels of F4/80. SPMs, a minor subset in unstimulated mice, have a F4/80(low). Infection or inflammation reduces LPMs levels and SPMs become the prevalent population (Bortoluci K.R. et al, Front Immunol, 2015). Moreover expression of Nef shifts M2 anti-inflammatory macrophage phenotype to partly pro-inflammatory M1 phenotype (Chihara 2012 J Immunol). In transgenic mice Nef expression reduces F4/80 staining and increases proportion of F4/80med/Mac-1-cells (Dickie P. 2000 AIDS Res Hum Retroviruses). We believe that while investigation of the macrophage subsets are interesting, addition of this large amount of data about macrophages phenotypes will complicate the story. Our main goal for the current study is test our hypothesis that macrophages migrate differently in lungs and peritoneal cavity. Due to complexity of lung macrophage phenotypes, we hope to investigate this in future.

Specific Comment 4. Fig 5: Inconsistency between A-E in the figure and A-D in the legend. 337-340: All these studies were performed in human primary macrophages or cell lines. What is the relevance for the Tg mouse macrophages? 

Response: We corrected inconsistence between the figure legend and the text.  All experiments described in the manuscript were performed with mouse peritoneal macrophages isolated either from WT or HIV-Tg mice. We didn’t use human primary macrophages in this study.

Specific Comment 5. Finally, I wonder what the mechanism is of “induced HIV expression” in this model. Any idea whether a specific viral transcript is responsible or whether it is just an effect of accumulation of any viral RNA, triggering RIG/NOD/TLR with type 1 IFN effector mechanisms. Has any attempt been done to delete particular HIV genes (e.g. Tat) in order to see if the effects are the same or different? This question may be relevant as to which additional therapeutic strategies should be developed to prevent co-morbidities: block any HIV transcription or more selective inhibition of e.g. Tat? 

Response: We have demonstrated previously that activities of CDK9/cyclin T1 and Sp1 are significantly reduced at 3% O2 comparing to higher activity at 21% O2. These changes were associated with increased levels of both Tat-activated and basal HIV-1 transcription (Charles S. 2009). Thus higher level of viral transcription leads to accumulation of viral proteins (tat, env, and Nef). We are currently investigating the role of Nef in the modulation of macrophages migration (paper under preparation). 

Reviewer 3 Report

1. Should provide rationale for not including HIV-Tg PBS treated control in experiment presented in Figure 1.

2. Number of mice per group should be included in Figure 1,2, 4, and 5 legends and/or individual animals shown on graphs.

3. Better explanation of the biological relevance / connection to oxygen tension for the specific levels of 5% and 21% used would be beneficial to include in the introduction, discussion, and/or results L205-210

4. PNMC, PMNC, PBMC should be clearly defined, particularly in reference to Figure 2.

5. L187.  Reference to Fig. 3A-D is confusing, are the authors referring to Fig. 3A-D in this manuscript or other manuscript?

6. L272-L274 and Figure 5A-C.  It appears in Fig 5A+B the histograms are being used to convey % of F48/80+ macrophages, but no gate is shown on the plots. Additionally, a p value (p=3.18x10-5) is provided in reference to Fig 5B, but unclear how that was determined.

7. Unclear if one-tailed or two-tailed t-tests were used.

Author Response

Specific Comment 1. Should provide rationale for not including HIV-Tg PBS treated control in experiment presented in Figure 1.

Response: We revised Figure 1 and added HIV-Tg-PBS treated controls as requested.

Specific Comment 2. Number of mice per group should be included in Figure 1,2, 4, and 5 legends and/or individual animals shown on graphs.

Response: We agree and added number of mice to each graph.

Specific Comment 3. Better explanation of the biological relevance / connection to oxygen tension for the specific levels of 5% and 21% used would be beneficial to include in the introduction, discussion, and/or results L205-210

Response: While the air oxygen level is 21%, it is much lower in body varying between 14% (lung) and 1% (bone marrow). Therefore, HIV-1 virus is exposed to the different oxygen levels in the organism. Expression of host factors essential for HIV-1 replications depends on the oxygen level and associated with oxygen sensor proteins HIF-1α and HIF-2. Most virological studies have been performed at 21% oxygen levels and the knowledge of the mechanistic aspects of viral infection is incomplete. HIF‐1α is stabilized at low oxygen conditions and can reduce NF‐κB activity in vivo and in vitro under inflammatory conditions (Bandarra D, et al, 2015 Dis Model Mech). Accordingly, we have demonstrated previously that activities of CDK9/cyclin T1 (the major host kinase involving in HIV-1 transcription) and transcriptional factor Sp1 are higher at 21% O2 compared to 3% O2 which are associated with increased levels of both Tat-activated and basal HIV-1 transcription (Charles S. 2009). Thus increased HIV-1 transcription can induce accumulation of viral proteins and escalate viral toxicity in organ specific manner. We added this information to the discussion.

Specific Comment 4. PNMC, PMNC, PBMC should be clearly defined, particularly in reference to Figure 2.

Response: We corrected abbreviation in the text and figure legends.

Specific Comment 5. L187.  Reference to Fig. 3A-D is confusing, are the authors referring to Fig. 3A-D in this manuscript or other manuscript?

Response: It was a typo and we corrected it.

Specific Comment 6. L272-L274 and Figure 5A-C. It appears in Fig 5A+B the histograms are being used to convey % of F48/80+ macrophages, but no gate is shown on the plots. Additionally, a p value (p=3.18x10-5) is provided in reference to Fig 5B, but unclear how that was determined.

Response: We added gates to the histogram and corrected text. We also update the statistical methods section to indicate that we used Students t-test for pairwise analysis, section 2.8.

Specific Comment 7. Unclear if one-tailed or two-tailed t-tests were used.

Response: Two tailed t tests were always used. We added this information to the Materials and Methods: Statistical analysis section 2.8.

Round 2

Reviewer 1 Report

The authors have addressed all of my concerns.

Author Response

We thank the reveiwer for the favorable decision.

Reviewer 2 Report

The authors performed a thorough revision of their text, with many clarifications. I could not find, however a point-by-point rebuttal. With regard to my remarks, I observe the following:

  • 126: How was confluence proven: just by microscopic view or by testing permeability to e.g. fluorescent beads of a certain size?

→ Has been correctly addressed in lines 138-143

  • Do the authors have any idea what the biological significance is of the two macrophage populations and the selective persistence of the ‘high F4/80” population after 72 H? Was there any attempt to separate and study both populations? Is this phenomenon related to proliferation or to functional alterations ?

→ I could not find an answer to these questions

  • Fig 5: Inconsistency between A-E in the figure and A-D in the legend.

→ This issue seems resolved

  • All these studies were performed in human primary macrophages or cell lines. What is the relevance for the Tg mouse macrophages?

→ Has been addressed in the Discussion

  • Finally, I wonder what the mechanism is of “induced HIV expression” in this model. Any idea whether a specific viral transcript is responsible or whether it is just an effect of accumulation of any viral RNA, triggering RIG/NOD/TLR with type 1 IFN effector mechanisms. Has any attempt been done to delete particular HIV genes (e.g. Tat) in order to see if the effects are the same or different? This question may be relevant as to which additional therapeutic strategies should be developed to prevent co-morbidities: block any HIV transcription or more selective inhibition of e.g. Tat?    

→ I could not find an answer to these questions.

During the reading of the present manuscript, I have the following additional remarks and questions:

  • 159: typo: Prism (not Prizm)
  • Fig 2B presumably shows the isotypic controls (see line 192 and 200), but that is not reflected in the labelling of the x and y axes.
  • For this experiment, it is not clear how absolute numbers were obtained, based on flow cytometry: i.e. were fluorescent beads used or were the cells microscopically counted? It is also not explicitly stated how gate P1 was defined: presumably based on FSC and SCC in Fig 2A?
  • 242: ends with “cultured at 3% O2.” While Fig 3F states culture at 5 % O Typo?

Moreover, I wonder why the authors were looking at PMNC at early, but not at later time points, because, obviously, those cells could also have a role in (persistence of) inflammation.

Author Response

Comment 1. Do the authors have any idea what the biological significance is of the two macrophage populations and the selective persistence of the ‘high F4/80” population after 72 H? Was there any attempt to separate and study both populations? Is this phenomenon related to proliferation or to functional alterations ?

→ I could not find an answer to these questions

Response: The levels of F4/80 expression are varying in macrophage subsets and during M1/M2 differentiation. M1 infiltrating macrophages express low levels F4/80 as has been demonstrated in WT infiltrating macrophages at 24 h (Fig.2F). It was surprising that at an early time point we found two populations of macrophages in HIV-Tg mice. These two populations may represent infiltrating macrophages with low levels of F4/80 and resident peritoneal macrophages expressing higher level of F4/80. Also, it is possible that HIV-1 gene expression might facilitate increased F4/80 levels in the infiltrating macrophages. The subset of macrophages expressing higher F4/80 levels increases at later time point (72 hrs) post LPS injection. This expansion might be associated with the increased macrophage proliferation. To fully translate mouse results into human, a future study is needed to analyze and characterize human macrophages infected with HIV-1 under the condition of low oxygen levels.

Comment 2. Finally, I wonder what the mechanism is of “induced HIV expression” in this model. Any idea whether a specific viral transcript is responsible or whether it is just an effect of accumulation of any viral RNA, triggering RIG/NOD/TLR with type 1 IFN effector mechanisms. Has any attempt been done to delete particular HIV genes (e.g. Tat) in order to see if the effects are the same or different? This question may be relevant as to which additional therapeutic strategies should be developed to prevent co-morbidities: block any HIV transcription or more selective inhibition of e.g. Tat?

→ I could not find an answer to these questions.

Response: We observed higher levels of tat, nef and env mRNA by qRT-PCR in lung macrophages compared to peritoneal macrophages in vivo. We also observed higher levels of these gene expression in peritoneal macrophages cultured at 21% oxygen compared to the peritoneal macrophages isolated from the same mouse but incubated at 5% oxygen in vitro. In this mouse model, there is an accumulation of viral mRNA and likely viral proteins, but not viral particles that are not produced in transgenic mice. Also this model only recapitulates transcription and post-transcription HIV-1 events as there are no early HIV-1 life events in this model. Thus RIG-I/NOD/TLR recognition and IFN signaling are not likely to be involved in the activation of HIV-1 transcription.  Activation of HIV-1 transcription requires host cell factors. We demonstrated previously that both basal and Tat-activated transcriptions were elevated in T cells and macrophages cultured at high oxygen (21%) (Charles at al., 2009). Upregulation of Tat- activated transcription was associated with the increased activity of CDK9/cyclin T1. Upregulation of basal HIV-1 transcription was associated with the increase of NF-kB. While murine cyclin T1 has a cysteine mutation that makes it inefficient in binding HIV-1 Tat, expression of Tat in murine epithelial cells induces HIV-1 transcription by several fold, comparing to 10-20 fold activation in human cells. This could be due to the cooperation of Tat with NF-kB or recruitment of histone acetyl transferases by Tat to HIV-1 LTR. Deletion of tat gene in all HIV-Tg models led to significant reduction of HIV-1 transcription. Moreover, we used 1E7-03 that prevents Tat binding to host PP1, the essential factor in HIV-1 transcription, and inhibits Tat-activated transcription.  Because Tat doesn’t possess enzymatic activity, allosteric inhibitors that reduces it’s binding to host factors are best specific inhibitors for Tat.

Comment 3. 159: typo: Prism (not Prizm)

Response: We corrected the typo

Comment 4. Fig 2B presumably shows the isotypic controls (see line 192 and 200), but that is not reflected in the labelling of the x and y axes.

Response: We corrected the Fig.2B

Comment 5. For this experiment, it is not clear how absolute numbers were obtained, based on flow cytometry: i.e. were fluorescent beads used or were the cells microscopically counted? It is also not explicitly stated how gate P1 was defined: presumably based on FSC and SCC in Fig 2A?

Response: The total number of cells isolated from abdominal cavity of each mouse was counted using Trypan blue assay. The gating strategy for P1 was used to exclude debris that have low forward scattering levels and include most cells with large size and granularity.

Comment 6. 242: ends with “cultured at 3% O2.” While Fig 3F states culture at 5 % O Typo?

Response: We corrected the typo to 5% oxygen (revised lane 243).

Comment 7. Moreover, I wonder why the authors were looking at PMNC at early, but not at later time points, because, obviously, those cells could also have a role in (persistence of) inflammation.

Response: We did not observe any difference in the number of peritoneal PMNC between WT and HIV-Tg mouse (Fig 1. F and Fig.2 E, H) during the early time points. Because infiltrating macrophages negatively regulate the following PMNC infiltration,  the levels of peritoneal PMNC in HIV-Tg mice were lower than in WT mice and unlikely play a major role in persistent inflammation.